# A Very Unusual Case of Physical Disability after Spinal Epidural Hematoma in the Course of Sport-Related Head Injury

Jan Gnus [1,2,*], Sebastian Fedorowicz [1], Karolina Radzikowska [1] and Anna Kołcz [1,3]

1  Department of Physiotherapy, Wroclaw Medical University, 50-368 Wroclaw, Poland
2  Regional Specialist Hospital in Wroclaw, Research and Development Center, 51-124 Wroclaw, Poland
3  Laboratory of Ergonomics and Biomedical Monitoring, Wroclaw Medical University, 50-368 Wroclaw, Poland
*  Correspondence: jan.gnus@umw.edu.pl

**Abstract:** Spinal epidural hematoma (SEH) is blood accumulation between the dura mater of the spinal canal and the bone of the vertebrae. It is estimated to be an extremely rare incidence, affecting approximately 0.1 of 100,000 patients. When the suspected cause is a sport-related injury in the majority of cases it affects the trauma region. The aim of this case report was to outline the case of a 60-year-old man who was admitted to the Emergency Department due to acute, severe pain in the lumbar region which lasted 30 min. It occurred that 54 h prior to the admission, he suffered a head injury due to sport-related trauma during recreational skiing. When waiting for the laboratory and imaging test results the patient developed bilateral paralysis of the lower limbs. The MR examination results showed SEH at the level Th9-L1; therefore, immediate neurosurgical laminectomy was performed. After 36 months of rehabilitation, the patient can walk independently. In summary, SEH without immediate and appropriate treatment is linked with very poor clinical outcome. Moreover, a high mortality rate of 7% and the fact that more than half of patients do not return to full physical health incentives its inclusion in differential diagnosis every time when symptoms of spinal cord involvement are found. Furthermore, delayed aftermath in the form of paresis of the lower limbs requires long-term and intensive physical rehabilitation.

**Keywords:** spinal epidural hematoma; head injury; sport-related trauma; paralysis; laminectomy; physical disability; rehabilitation; case report

## 1. Introduction

Spinal epidural hematoma (SEH) is a relatively rare disease, defined as blood accumulation in the epidural space, between dura mater and spinal canal, which results in increasing compression of spinal cord. As a result acute symptoms may occur, such as pain or serious neurological deficits [1]. Its incidence is estimated at 1:1,000,000 patients yearly accounting for less than 1% lesions related to the epidural space [1].

Misdiagnosis of SEH or delayed implementation of treatment results in serious consequences [2]; therefore, knowledge of SEH pathomechanism and its risk factors seems to be essential for enabling fast diagnosis to clinicians, even at the stage of medical history interview.

In terms of pathomechanism, SEH can be iatrogenic, non-iatrogenic or idiopathic. Iatrogenic is an effect of lumbar puncture and antiplatelet therapy; non-iatrogenic occurs after injury in patients with congenital coagulation disorders and in pregnant women; and idiopathic SEH can be of unknown origin [3,4].

The most common cause of SEH (22.2%) is trauma or sport-related injury with the hematoma in the region of the trauma [3,5]. Activities that in multiple cases led to SEH are: stretching exercises, daily work with physical activity, spinal chiropractic therapy, sportive training such as weight lifting, performing sit-ups, high-intensity swimming, breath-hold diving. As a consequence, vertebral and ligaments damage, excessive tension and extension of the trauma region may occur [6].

In the paper, the case of a man is presented, who experienced craniocerebral trauma and 54 h after the injury, there were features of the epidural hematoma of the spinal cord. In the literature review performed by us, we have not found similar coincidence. The review included English literature of the subject published up to May 2020 using Medline and PubMed Central search engines, using the following MeSH keywords: "spinal epidural hematoma", "spinal epidural hemorrhage", "head injury", "head trauma", "mild brain injury".

## 2. Case Report

A 60-year-old man was admitted to the Emergency Department due to acute, severe pain in the lumbar region which lasted 30 min. More than two days prior to the admission (exactly 54 h before) during recreational skiing, the patient experienced head damage with an eyebrow and right arm injury due to being pushed by another slope participant and falling to the right side. The patient denies loss of consciousness. The eyebrow injury had been dressed independently by the patient, who had been felt dizziness, vertigo and head pain, accompanied by nausea and well-being deterioration, which all resolved spontaneously. Three hours after the accident and after an improvement of well-being, the patient continued physical activity.

In the physical examination, bilateral soreness of the lumbar region had been stated, Goldflam's sign was difficult to assess. In the neurological examination, bilateral impairment of the lower limbs' muscle strength was rated using the Medical Research Council (MRC) Scale and rated at 4/5 with maintaining sensory functions and sphincter functions. Due to severe pain issues, metamizole and drotaverine had been administered in a drip infusion while waiting for the laboratory and imaging tests results.

Within a short time after infusion absorption, the patient developed paralysis of the lower limbs. In another examination, muscle strength had been assessed at 0/5 in the MRC scale in both limbs. Loss of touch sensation, pain, temperature and sphincter dysfunction have been stated. In neurological examination of upper limbs, no defects were found. Laboratory results, including coagulation system, have been correct. The following comorbidities and conditions were noted: ischemic heart disease, a history of the entire-wall myocardial infarction, and kidney stones. The following medications are taken: 75 mg of aspirin.

In the computed tomography (CT) examination, no alarming signs indicating the presence of intracranial hematoma, edema or contusion of the brain as well as skull fractures were confirmed. At the same time, in the performed magnetic resonance imaging (MRI) examination of thoracic and lumbar spine (Figure 1) at the Th9-L1 level, epidural hematoma with a width up to 1.4 cm and dorsal localization was noted. Moreover, on both sides there are visible signs of compression and spinal cord edema at this level. The change signal was heterogeneous in all sequences, corresponding to the presence of hematoma, forming three fluid spaces with the largest of dimensions being 2.2 cm × 1.0 cm. The signal from the caudal segment was elevated, which indicated spinal cord edema.

After obtaining the results of the imaging tests, the patient was transferred to the Neurosurgical Department. Due to the MRI signs and serious neurological deficits, the patient was qualified for surgical treatment. Laminectomy had been performed at the height Th9-Th12 with hematoma evacuation and decompression of the spinal cord. In the MRI examination performed after laminectomy, features of the enlargement and swelling of the spinal cord are visible at the section about 8.5 cm with a narrowing of the spinal cord, with moderately expanded fluid space at the level of executed intervention (Figure 2).

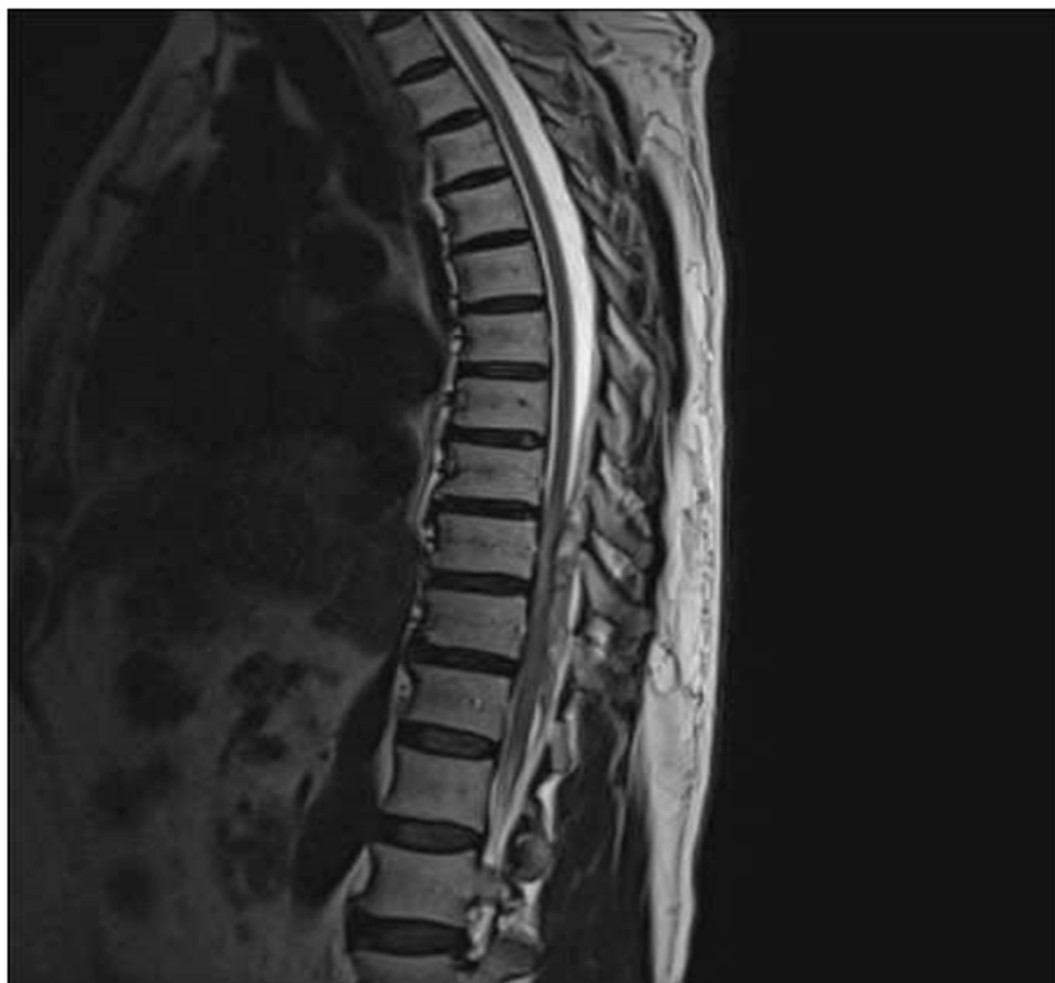

**Figure 1.** MRI scan of the spine was performed in SE sequences of T1 and T2-dependent images, TIRM and SWI. MRI was performed at the level from Th2 to L4 [17 February 2017]. At the Th9-L1 level, ESH is visible dorsally, with a width of up to 1.4 cm. Moreover, there are visible signs of compression and spinal cord edema at this level. The change signal was heterogeneous in all sequences, corresponding to the presence of hematoma, forming three fluid spaces with the largest of dimensions 2.2 cm × 1.0 cm. The signal from the caudal segment of the spinal cord was elevated, which indicated spinal cord edema.

In the postoperative course, a slight improvement in superficial and deep sensation was obtained, without improvement in the motor function of the lower limbs (MRC 0/5). The patient was referred to the rehabilitation ward, where he underwent therapy under the concept of proprioceptive neuromuscular facilitation (PNF).

As a result of long, intensive rehabilitation after 6 months, the patient regained control of the sphincter function, and after 9 months the patient made the first lower limbs movements. After 18 months of rehabilitation, motor function in the lower limbs improved on both sides to 4.5/5 at MRC grade, and the full range of sensory function returned. Currently, after 36 months of rehabilitation, a significant improvement in the motor functions of the lower limbs has been obtained. However, disorders in the strength of the biceps femoris muscle and quadriceps femoris persist. The patient is able to move independently, with the help of a walker and orthopedic crutches.

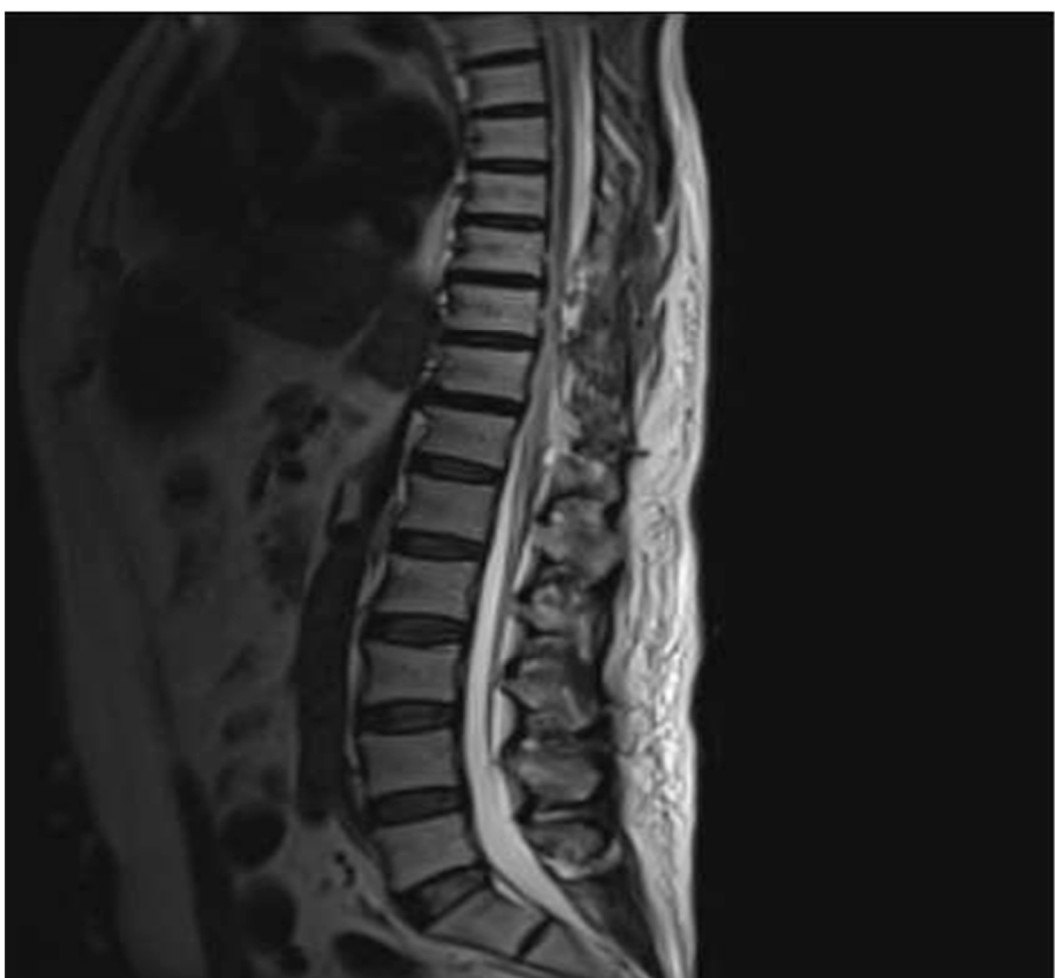

**Figure 2.** Condition after the Th9-Th12 laminectomy and after removal of ESH [21 February 2017]. MRI examination was performed in SE sequence, T1 and T2 dependent images and in GRE FS sequence. The scope of the examination was Th6-S3. Features of the enlargement and swelling of the spinal cord at a section of about 8.5 cm with a narrowing of the spinal cord. Moderately expanded fluid space at the laminectomy level. In addition, the thoracic spine and thoracic spinal cord do not show changes in MRI.

### 3. Discussion

ESH is a disease encountered in clinical practice extremely rarely—its prevalence in the population is estimated at 1:1,000,000 [2]. However, taking into account the high mortality rate of 7% [3] and the fact that more than half of patients do not return to full physical health (8), SEH without doubts should be included in the differential diagnosis in each case with suggestive symptoms of spinal cord involvement. The presence of SEH risk factors in clinical history seems to be a valuable diagnostic clue.

The first historical mention of SEH appeared in the literature in 1869 thanks to R. Jackson [7], since then more than 1000 cases have been described [8], although the exact mechanism of hematoma formation is not clear and various theories have been appearing for years to explain this issue. Most authors agree that the internal vertebral plexus is the main anatomical structure that causes bleeding [9]. It is particularly susceptible to this because of its valvular structure that forms the anastomosis chain from the base of the skull to the sacrum. Therefore, this entire network remains sensitive to pressure changes in the abdominal cavity or chest [10]. Numerous cases of SEH called specifically spontaneous SEHs have been reported in the literature in link with coughing, sneezing, strenuous

physical exercise like weight lifting, sustained Valsalva maneuver or even while playing a wind instrument [11].

However, the theory that SEH can be caused by arterial bleeding [12] also finds support in the literature. Some authors believe that the time at which the symptoms develop indicates the source of origin—arterial bleeding manifests itself by the acute onset of symptoms, while bleeding from the venous plexus, due to the low pressure prevailing there, causes a slower growth of the hematoma and gradual disclosure of symptoms [8].

In the case described above, pain and paresis progressed extremely rapidly, suggesting arterial bleeding. Nonetheless, it did not agree with the history of the injury two days prior to the admission, nor with the picture of hematoma on the MRI. Due to evolution of hematoma's magnetic properties over time, it is possible to estimate its formation time [13]. Hematomas formed in less than 24 h are isointense in the T1 sequence and iso- or hyperintense in T2-dependent images. Subacute hematomas formed within 24–72 h remain isointense in T1-dependent images, while due to oxygen dissociation from hemoglobin and deoxyhemoglobin formation, the T2-dependent image begins to be heterogeneously hypointense [14]. This is the picture we have obtained of our patient, which allows us to conclude that the hematoma has been slowly growing since the injury, probably as a result of venous bleeding from the internal vertebral plexus.

Despite the proven connection of the venous system of the skull cavity and the internal vertebral venous plexus and the presence of head trauma in the medical history, after analyzing the published literature it seems that the direct cause of bleeding into the epidural space is not head trauma. This is indicated by statistical data and pathophysiology of changes taking place after head injury. There are reports of increased blood flow through the internal vertebral venous plexus after concussion; however, it is a long-term process, resulting from several weeks of structural changes in the brain's microcirculation, which does not occur during the injury itself. Moreover, with a change of body position from vertical to horizontal, the outflow of blood from the skull cavity through the internal jugular veins increases, reducing and relieving the flow through the internal vertebral plexus. In addition, our patient has other risk factors for SEH, such as antiplatelet therapy and fall dynamics itself, which results in a sudden increase in chest pressure.

Therefore, in this case, the etiology of hematoma seems to be multifactorial, and it is unclear how head injury is involved in the pathogenesis of she. In the light of the cited data, a relationship between SEH and head injury alone, without other risk factors, seems unlikely but it cannot be ruled out.

## 4. Conclusions

The late consequences of craniocerebral trauma very rarely causes the formation of an epidural hematoma of the spinal cord, especially in a different area than the one affected by the injury. Rapid diagnostics and surgical treatment increase the chances of a good prognosis. Delayed aftermath in the form of paresis of the lower limbs require long-term and intensive physical rehabilitation.

**Author Contributions:** Conceptualization: J.G., S.F., K.R. and A.K.; methodology: J.G. and S.F.; software: K.R.; validation: A.K.; formal analysis: S.F. and K.R.; investigation: S.F. and K.R.; resources: A.K.; data curation: K.R. and A.K.; writing—original draft preparation: S.F. and K.R.; writing—review and editing: J.G. and A.K.; visualization: S.F. and K.R.; supervision: J.G. and A.K.; project administration: J.G.; funding acquisition: J.G. and A.K. All authors have read and agreed to the published version of the manuscript.

**Funding:** This case study is a part of larger research project was funded by the Ministry of Science and Higher Education of Poland under the statutory grant of the Wroclaw Medical University (no. SUBZ.E060.22.099).

**Institutional Review Board Statement:** The study was conducted in accordance with the Declaration of Helsinki, and approved by the Bioethics Committee of the Wroclaw Medical University.

**Informed Consent Statement:** Informed consent was obtained from the subject involved in the study.

**Data Availability Statement:** The data presented in this study are available on request from the corresponding author.

**Acknowledgments:** There were no other contributors to the article than the authors as well as there was no writing assistance regarding our paper. We would like to thank the patient for his contribution in this research.

**Conflicts of Interest:** The authors declare no conflict of interest.

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
