# Peer review of "A Very Unusual Case of Physical Disability after Spinal Epidural Hematoma in the Course of Sport-Related Head Injury"

_sustainability, doi:10.3390/su142215409_

Round 1
Reviewer 1 Report
In the manuscript presented by Jan Gnus et al., the authors presented a case of spinal epidural hematoma after sport-related head injury. This case is very rare and the mechanism of hematoma formation is not yet elucidated. The authors also discussed the likely etiology of hematoma related with head injury. The presented case is interesting and the discussion is well-written. However, there are a few issues that need to be addressed before publication.
1. In case report presentation, the medical history of ischemic heart disease, myocardial infarction, kidney stones, and aspirin taking should be described after the recent history of head damage.
2. Though no direct injury to the spine was seen, physical examination should exclude bruise of skin of the back. Is there any possibility that the epidural hematoma started to form at the time point of the accident?
3. Did the patient underwent head CT or MRI scan? Were there signs of intracranial hematoma, edema or contusion on the imaging?
Author Response
REVIEWER 1
In the manuscript presented by Jan Gnus et al., the authors presented a case of spinal epidural hematoma after sport-related head injury. This case is very rare and the mechanism of hematoma formation is not yet elucidated. The authors also discussed the likely etiology of hematoma related with head injury. The presented case is interesting and the discussion is well-written. However, there are a few issues that need to be addressed before publication.
- Reply: We appreciate such motivating feedback; it means a lot to us. All suggestions have been addressed and the manuscript has been improved accordingly.
- In case report presentation, the medical history of ischemic heart disease, myocardial infarction, kidney stones, and aspirin taking should be described after the recent history of head damage.
- Reply: As suggested, this part of the manuscript has been improved and the sentences pointed by the Reviewer have been relocated from the paragraph 1 in Case Report section to the paragraph 3 (lines 79-82): “The following comorbidities and conditions were noted: ischemic heart disease, a history of the entire-wall myocardial infarction, and kidney stones. The following medications are taken: 75 mg of aspirin.”
- Though no direct injury to the spine was seen, physical examination should exclude bruise of skin of the back. Is there any possibility that the epidural hematoma started to form at the time point of the accident?
- Reply: Thank you for this important opinion. We would like to clarify that no unilateral or bilateral bruisers were noted in the physical examination. Patient clearly reported that during the accident and head injury, he fell to the right side of the body.
- Did the patient underwent head CT or MRI scan? Were there signs of intracranial hematoma, edema or contusion on the imaging?
- Reply: Yes, patient underwent CT examination with no alarming signs indicating the presence of intracranial hematoma, edema or contusion of the brain as well as skull fractures. This information was provided in the paragraph 4 of the Case Report section (lines 83-85): “In the computed tomography (CT) examination, no alarming signs indicating the presence of intracranial hematoma, edema or contusion of the brain as well as skull fractures were confirmed.”
The remaining issues pointed by the Reviewer 1 in the checklist report have been improved, namely: research methods and discussion of findings.

Reviewer 2 Report
In my opinion this case reoprt is sutibale for publication
Author Response
REVIEWER 2
In my opinion this case reoprt is sutibale for publication.
- Reply: We are grateful for the acceptance of our manuscript in the present form. The Reviewer’s opinion is really important ang gives us motivation for further research.
There were no remaining issues from the Reviewer 2 based on the checklist report to be improved.

Reviewer 3 Report
A clear case study has been presented by the authors, it could be published
Author Response
REVIEWER 3
A clear case study has been presented by the authors, it could be published.
- Reply: We are grateful for the acceptance of our manuscript in the present form. The Reviewer’s opinion is really important ang gives us motivation for further research.
The remaining issues pointed by the Reviewer 3 in the checklist report have been improved, namely: description of the content based on the theoretical background and empirical research on the presented topic, relevant references, and discussion of findings.

Reviewer 4 Report
Spinal epidural hematoma is not that rare.
Author Response
REVIEWER 4
Spinal epidural hematoma is not that rare.
- Reply: SEH with typical spinal etiology and pathomechanism is not rare, indeed. However, we talk about SEH in the course of head injury with no spinal injury and this has not been explained so far. Our opinion is in line with the Reviewer 1 comment: “The authors presented a case of spinal epidural hematoma after sport-related head injury. This case is very rare and the mechanism of hematoma formation is not yet elucidated. The authors also discussed the likely etiology of hematoma related with head injury.” Nevertheless, we appreciate this respectful opinion and we thank for accepting this paper for publication.
The remaining issues pointed by the Reviewer 4 in the checklist report have been improved; namely: description of the content based on the theoretical background and empirical research on the presented topic, discussion of findings, and conclusions based the obtained results and with reference to the previous studies; as well as relevant references, research methods, and empirical results.

Round 2
Reviewer 4 Report
As I wrote on the previous review, this manuscript has nothing new to report. it is not that rare in neurology/neurosurgery.